# The Quality of Air in Polish Health Resorts with an Emphasis on Health on the Effects of Benzo(a)pyrene in 2015–2019

Ewa Anioł [1,*], Jacek Suder [2], Jan Stefan Bihałowicz [3] and Grzegorz Majewski [1]

1   Faculty of Civil and Environmental Engineering, Warsaw University of Life Sciences SGGW-WULS, 02-787 Warsaw, Poland; grzegorz_majewski@sggw.edu.pl
2   Faculty of Safety Engineering and Civil Protection, The Main School of Fire Service, 01-629 Warsaw, Poland; 12835@student.sgsp.edu.pl
3   Institute of Safety Engineering, The Main School of Fire Service, 01-629 Warsaw, Poland; jbihalowicz@sgsp.edu.pl
*   Correspondence: ewa_aniol@sggw.edu.pl

**Abstract:** The aim of this paper was to analyze the impact of air pollution and meteorological conditions on the effectiveness of recreation in selected health resorts in Poland in 2015–2019. Four municipalities with the status of health resorts were compared in terms of exposure to harmful air pollutants such as $PM_{10}$, $NO_2$, $SO_2$, and $B_{(a)}P$ in $PM_{10}$. In this paper, a comprehensive statistical analysis was performed by determining the basic statistics of the measurement series. In addition, analyses of the occurrence of episodes of elevated $PM_{10}$ concentrations in health resorts in Poland, as well as correlation and regression analyses, were performed. Statistical analysis showed no annual mean exceedances for the air pollutants analyzed. Average annual concentrations of harmful pollutants decreased year by year in Rabka Zdrój and also in Ciechocinek. The situation was different in Sopot and Ustroń, where the average annual pollution remained at a similar level and there was no downward trend. Studies have shown that travel to spa communities for health purposes can be problematic because air quality, while not exceeding average annual standards, is not satisfactory. To effectively address public health concerns, it is also necessary to consider meteorological conditions when analyzing air quality. A detailed analysis of the impact of meteorological conditions (average air temperature, wind speed, relative humidity, and visibility) on air quality, based on forecasts, will also help in the implementation of air protection plans and strengthen the control of harmful pollutant levels. Measures to reduce the levels of harmful pollutants will affect the effectiveness of patient treatment in spas. The article presents the correct way to conduct reliable monitoring of air quality and meteorological conditions, where it is particularly important.

**Keywords:** air quality; air pollution; benzo(a)pyrene; health resorts

## 1. Introduction

Air pollution, as a result of globalization and human activity in the environment, is now a common phenomenon that poses a serious threat to human health and life. The level of air purity in increasingly large Polish cities is being significantly reduced, generating an increase in threats to citizens' health security [1]. Air quality is not only a factor that directly and indirectly affects health, but also an important factor in determining the quality of life (wellbeing). Individual social groups may be more affected by air pollution [2]. Elderly people or families with children who spend active time outdoors are more likely to be exposed to the negative effects of harmful pollutants. The use of statistical methods has shown that air pollution has a significant link with mental wellbeing [3]. Individual statistical analyses may allow air pollution reports to be implemented into public healthcare and risk communication. This is important both for health resorts and for the general public, particularly residents of urban agglomerations. The status of health resort is granted by the municipality, within the limits of documented resources of natural medicinal raw materials

(healing waters and peloids) and favorable climate properties. The healing properties of the climate are determined by atmospheric factors that promote the preservation of health, as well as the treatment and relief of the effects and symptoms of diseases [4–7].

In accordance with the current law [8], the status of health resort may be granted to an area that together satisfies the following conditions:

(1)  has deposits of natural medicinal raw materials (gaseous or fossil) with proven medicinal properties;

(2)  has a climate with medicinal properties (atmospheric factors conducive to the preservation of health, as well as the treatment or alleviation of the effects or symptoms of diseases);

(3)  has adequate facilities (hospitals, sanatoriums, and natural medical facilities, as well as drinking rooms, teetering, parks, movement paths, decorated sections of the sea coast, healing and rehabilitation health resort pools, and decorated underground mining excavations) for treatment relative to its purpose;

(4)  meets environmental requirements;

(5)  has technical infrastructure for water, wastewater and energy management, and public transport, as well as carries out appropriate waste management, including medical waste management.

Health is the most important value for a person; however, in recent years, there has been a perceived negative effect of air pollution on human health [1]. In Poland, air pollution with $PM_{10}$, $PM_{2.5}$, and benzo(a)pyrene is a problem. Their most important sources are low emissions, i.e., exhaust gases from boilers and solid fuel furnaces in households, as well as particulate matter, road transport, and energy [9]. The link between the rapid increase in the number of visitors and the higher levels of $PM_{10}$ is due to the fact that regional pollution is affected by both meteorological and anthropogenic emissions [10].

The phenomenon of low emissions observed in health resorts, which, due to geographical conditions, usually occurs from October to the end of May, poses a serious threat to the health of the inhabitants of the municipality, as well as questions the legitimacy and effectiveness of the treatment offered during periods of increased air pollution by carcinogenic particulate matter ($PM_{10}$, $PM_{2.5}$), nitrogen oxides, sulfur dioxide, or polycyclic aromatic hydrocarbons (PAHs), including the particularly dangerous benzo(a)pyrene [11]. Polycyclic aromatic hydrocarbons are commonly found in the environment. They are formed during the incomplete combustion and pyrolysis of organic substances of both natural and anthropogenic origin. Polycyclic aromatic hydrocarbons, upon penetrating the skin or mucous membrane, spread throughout the body. They localized in all tissues containing fat, and they are stored in adipose tissue, kidneys, liver, and lungs, from which they are gradually released [12]. Numerous studies have comprehensively documented the increased risk of lung cancer due to exposure to PAHs in a specific environment [13,14]. PAHs generally come from the incomplete combustion of organic matter, such as coal, oil, wood, and other fuels, through pyrolysis or pyrosynthesis [15,16]. However, a large share of atmospheric PAHs can be found in emissions from vehicles, especially diesel vehicles [17,18]. In addition to the above, coal burning, biomass kitchen stoves, particulate matter emissions, brick kilns, and smoking/cigarettes are other major sources of PAHs in the air [15,16]. Air pollution negatively affects living organisms by shortening life expectancy and increasing mortality. The adverse effects of certain air components have long been known. These include nitrogen oxides (NO), carbon dioxide ($CO_2$), and sulfur dioxide ($SO_2$), which are directly ingested into our body during the breathing process. Gaseous and particulate pollutants [19] have the same negative impact on human health. According to the World Health Organization (WHO), between six million and seven million people die prematurely from air pollution worldwide; there are approximately 40,000 to 45,000 deaths annually in Poland, about 70% of which are mainly due to heart and cardiovascular diseases [20]. This is why it is so important to take action to combat low emissions, especially in towns with health resorts, whose healing qualities are due not only to the specificity of the medicinal mines extracted, but also to the climate, which, it turns out, only meets

quality standards for a small part of the year [11]. The problem of poor air quality in Poland affects a large part of the country, including selected health resort areas [21], which should show a particularly high level of environmental and air quality [8]. Therefore, improving air quality, especially in health resort areas, can play a key role in the national strategy for reducing atmospheric air pollution, as well as in programs and actions implemented at the regional and local levels [22]. The main objective of this study was to determine the levels of harmful pollutants in selected cities for analysis, with reference to the results of the standards contained in the Regulation of the Minister of the Environment from 8 October 2019, which amended the Regulation on the levels of certain substances in the air [23]. The added value of the study is an assessment of cancer exposure with inhalation exposure to benzo(a)pyrene for selected cities with health resort status throughout Poland.

## 2. Materials and Methods

The study was based on 24 h data of $PM_{10}$, $NO_2$, and $SO_2$ concentrations and $B_{(a)}P$ levels from monitoring conducted by provincial environmental inspectorates in 2015–2019. Data on air pollution levels were taken from the Measurement Data Bank, and meteorological data were taken from ogimet.com [24]. Three monitoring stations such as Ustroń, Sopot, and Rabka Zdrój perform automatic measurements, while one station located in Ciechocinek performs manual and automatic measurements (Table 1). Due to a lack of measurements of meteorological conditions, data for the spa town (Figure 1) were taken from Bielsko-Biala, 30 km away. Results of meteorological measurements for Rabka Zdrój were taken from nearby Zakopane. Zakopane used to be a spa town, but it lost its status of a spa commune in 1966 [25]. Due to the lack of continuity of data and interruptions in measurements carried out for Ciechocinek, a health resort community located in the Kujawsko-Pomorskie Province, data on the level of harmful $SO_2$ pollution and meteorological conditions prevailing in this area were taken from Toruń, which is 30 km away. For Sopot health resort, data on meteorological conditions and $B_{(a)}P$ were collected from 12 km away in Gdańsk, which, together with Gdynia, is a part of Tricity.

**Table 1.** Basic characteristics of the analyzed pollution monitoring stations in selected health resorts.

| Health Resorts | Year of Receipt of the Statutes of the Health Resorts | Address | Environment | Φ | λ | Station Type | Measured Impurities |
|---|---|---|---|---|---|---|---|
| Ciechocinek | 1836 | Tężniowa St. Park Tężniowy | Also Park, recreational facilities | 52,888,422 | 1,878,091 | m/a | $PM_{10}$, $NO_2$, $NO_x$, $O_3$, benzene PAH, metals in $PM_{10}$ |
| Rabka Zdrój | 1864 | Sienkiewicza St. | Small meadow, around tree and residential development | 49,293,564 | 1,996,008 | a | $PM_{10}$, $PM_{2.5}$, $NO_2$, $NO_x$, $SO_2$, benzene |
| Sopot | 1823 | Bitwy pod Płowicami St. | Near Karlikowski's forest | 5,443,167 | 1,857,972 | a | $PM_{10}$, $NO_2$, $NO_x$, $SO_2$, CO |
| Ustroń | 1804 | 7 Sanatorium St. | Around sanatoriums | 4,971,973 | 1,882,672 | a | $PM_{10}$, $NO_2$, $NO_x$, $SO_2$, $O_3$ |

Station types: a—automatic; m—manual.

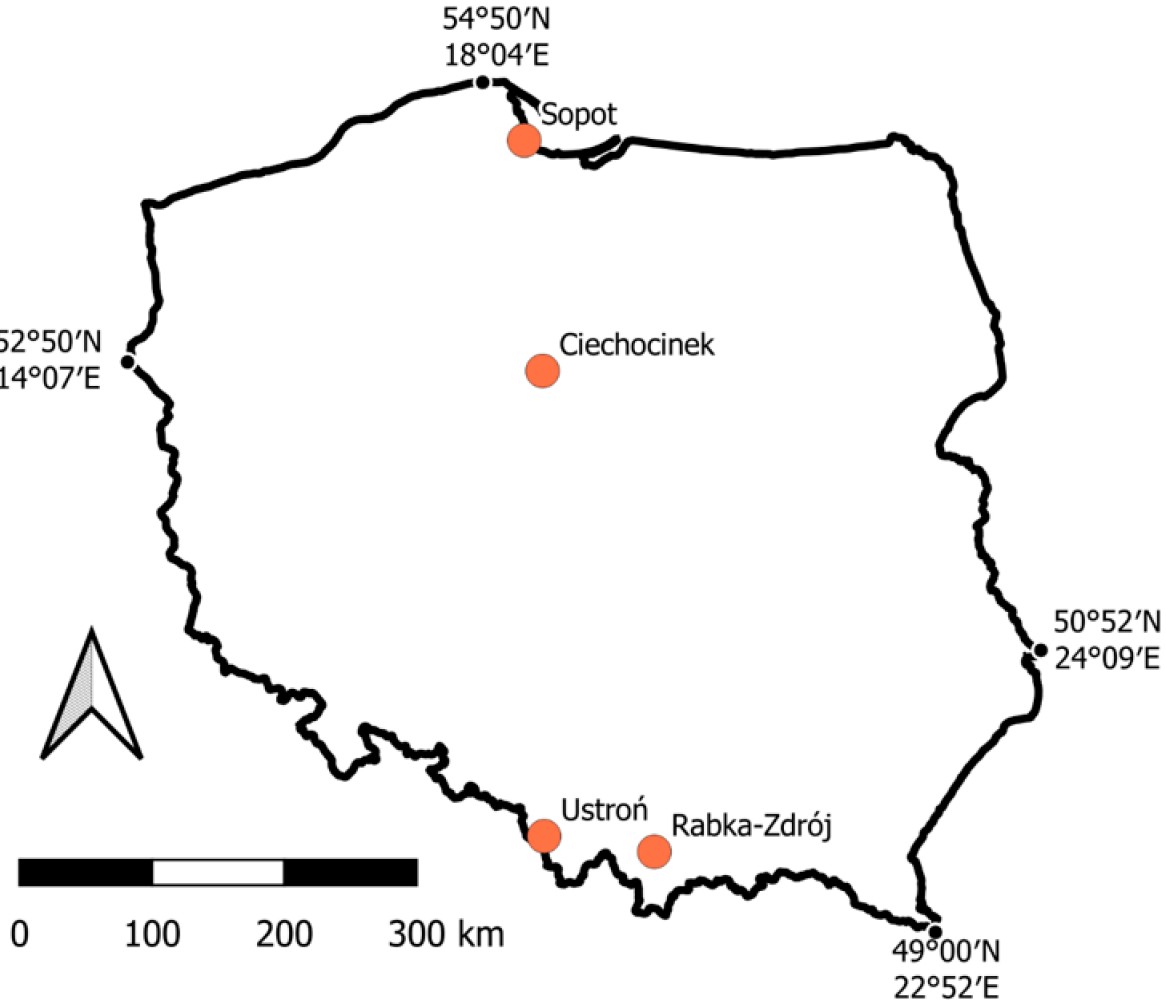

**Figure 1.** Location of studied health resorts.

The article provides statistical analyses such as:

- Analysis of the basic statistical characteristics of the measuring series;
- Analysis of the occurrence of episodes of elevated $PM_{10}$ concentrations in health resorts in Poland;
- Analysis of correlations between pollutant concentrations and meteorological parameters with visibility;
- Regression analysis between air pollution from meteorological and visibility tests;
- Calculation of the excess cancer risk according as a ratio of cancer risks calculated according to US EPA methodology.

To perform regression analysis, a step-back type is selected because it eliminates errors and is more accurate than step-progressive. We used the Pearson coefficient of correlation since all the data we used in the model met the assumptions of normality [26]. In the analysis the independent variable is visibility, and dependent variables are air pollutants such as $PM_{10}$, $NO_2$, $SO_2$, $B_{(a)}P$ in $PM_{10}$ and meteorological conditions: average air temperature $T_{av}$, wind speed $W_s$, and relative air humidity *RH*. The table annexed below provides basic information on the monitoring stations analyzed in selected health resorts. Table 1 specifies the type of stations, their location, and their surroundings. In addition, the types of impurities tested and the year of commencement of measurements are specified.

*Health Risks Associated with Benzo(a)pyrene*

Based on the U.S. EPA's benzo (a) pyrene cancer risk assessment methodology, we used excess cancer risk (*ECR*) as a measure comparing cancer risk (*CR*) at health resorts with reference sites.

In the risk assessment of the US EPA, the *CR* is defined as

$$CR = D \cdot SF_{\text{B(a)P}}$$

where $D\left[\frac{mg}{d\cdot kg}\right]$ is absorbed dose, and $SF_{\text{B(a)P}}$ the carcinogenic factor for $B_{(a)}P$. The dose is given by formula:

$$D = \frac{C \cdot K}{M}$$

where $C\left[\frac{mg}{m^3}\right]$ is mean concentration of $B_{(a)}P$, $K\left[\frac{m^3}{d}\right]$ is the mean daily ventilation, and $M[\text{kg}]$ is person body weight.

We define *ECR* as the difference of *CR* between a health resort $CR_{hr}$ and the reference site $CR_{ref}$ divided by risk at reference site: $CR_{ref}$

$$ECR = \frac{CR_{hr} - CR_{ref}}{CR_{ref}}$$

Assuming that we perform calculations for the person that has the same body mass at the reference site and in a health resort, and we compare the risk per year, the formula simplifies to a comparison of yearly average concentrations of $B_{(a)}P$

$$ECR = \frac{C_{hr} - C_{ref}}{C_{ref}}$$

The *ECR* can have values from $-1$ (if the concentration at health resort $C_{hr} = 0$) to positive, high numbers (for $C_{hr} \gg C_{ref}$).

In the analysis, the $C_{ref}$ will be the average yearly concentration of $B_{(a)}P$ in Poland based on [27].

## 3. Results

### 3.1. Basic Statistical Characteristics of the Measuring Series

The following are tables with basic statistical characteristics for health resorts such as Ciechocinek, Rabka Zdrój, Sopot, and Ustroń in 2015–2019. The analyzed meteorological parameters are: average air temperature, relative humidity, speed, and concentrations of air pollutants ($PM_{10}$, $SO_2$, $NO_2$,) and benzo(a)pyrene content in $PM_{10}$ suspended particulate matter recorded in 2015–2019.

According to the Minister's Regulation of 24 August 2012 on the levels of certain substances in the air, during the calendar year, the level of $PM_{10}$ particulate matter should not exceed the standard of 40 μg/m$^3$ [28]. On the basis of Table 2, it was shown that in Ciechocinek in 2015–2019, the average $PM_{10}$ particulate matter level standard was not exceeded. In recent years, due to unsatisfactory air quality, air protection programs have been carried out in Poland. In many cities such as Ciechocinek, the practical implementation of these programs results in lower levels of harmful pollutants. The aim of the establishment of air protection programs is to improve air quality and to comply with the standards set out in the Regulation in areas where exceedances occur. The practical implementation of air protection programs results in a reduction in the levels of harmful pollutants. A favorable trend has been noted, as from year to year, the average level of harmful pollutants gradually decreases. The same has been shown in Rabka Zdrój, where average annual concentrations of particulate matter have been reduced from 32.98 μg/m$^3$ in 2015 to 25.01 μg/m$^3$ in 2019. However, particularly untapped situations are in Sopot and Ustroń, where average annual particulate matter pollution is at a similar level and not decreasing. Previous studies

have shown that in conditions of long-term effects, there is most likely no safe level of air pollution below which adverse health effects are no longer observed [29]. The highest concentrations of $SO_2$ (Table 3) were observed in 2017 and 2018 in the health of Rabka Zdrój. Although the exceedance levels were insignificant, the air quality in the health resorts of Rabka Zdrój compared to Ciechocinek or Ustroń looks untapped. For example, in 2018, the concentration of $SO_2$ air pollution in Ciechocinek (2.63 μg/m$^3$) was 8 times lower than in Rabka Zdrój (20.46 μg/m$^3$). The most favorable situation was observed in the health resorts of Sopot, where the average annual concentrations of $SO_2$ in 2015–2019 did not exceed 2.00 μg/m$^3$, and a further decrease in these values was observed. Winds by the sea blow more often and harder than in other regions Polish, which effectively contributes to the reduction of concentrations of harmful pollutants. Analyses of the average annual concentration values for $NO_2$ sulfur dioxide (Table 4) indicate that, within five years, the annual standard value has not been exceeded which, in accordance with the Regulation of the Minister of 24 August 2012, is 40 μg/m$^3$. The most unfavorable situation—the highest annual average concentration occurred in Ustroń (Table 4) in 2017, followed by a sudden decrease in 2018 and gradually decreasing from 2019. $B_{(a)}P$ in $PM_{10}$ was found to decrease level in 2015–2019. Despite the steady decrease in $B_{(a)}P$ concentrations in the air since the 1990s, exceedances of its permitted level (1 ng/m$^3$) [28] are still recorded in the predominant area of the country. In 2005, under Directive 2004/107/EC of the European Parliament and of the Council of 15 December 2004 on arsenic, cadmium, mercury, nickel, and polycyclic aromatic hydrocarbons, an obligation was introduced in Poland to measure the benzo(a)pyrene content in particulate matter [30]. From the above data (Table 5), it appears that environmental pollution by PAHs compounds in Poland may pose a serious health risk. It therefore appears necessary to take all measures to reduce the exposure of the population to these compounds. According to data from the European Environment Agency, more than 46,000 people die every year from poisoned air in Poland. Achieving the required levels of reduction of particulate matter and benzo(a)pyrene emissions from the municipal and living sectors, at the current rate of action, can take 24 to almost 100 years on a per-state scale [31].

**Table 2.** Average annual values of $PM_{10}$ in the studied health resorts in 2015–2019.

| Health's Resort | Ciechocinek | | | Sopot | | |
|---|---|---|---|---|---|---|
| Year | Mean ± SD | Min | Max | Mean ± SD | Min | Max |
| 2015 | 27.00 ± 19.27 | 2.10 | 114.10 | 14.26 ± 8.36 | 3.80 | 68.60 |
| 2016 | 25.21 ± 15.68 | 2.80 | 96.90 | 16.74 ± 8.96 | 3.70 | 57.80 |
| 2017 | 24.61 ± 21.61 | 2.80 | 169.90 | 16.86 ± 13.10 | 3.20 | 108.30 |
| 2018 | 24.90 ± 16.71 | 4.10 | 87.80 | 21.54 ± 12.63 | 3.70 | 77.80 |
| 2019 | 21.18 ± 11.82 | 3.80 | 63.90 | 18.67 ± 9.94 | 3.00 | 53.00 |
| Health's Resort | Rabka Zdrój | | | Ustroń | | |
| 2015 | 32.97 ± 24.32 | 4.30 | 176.90 | 23.17 ± 12.61 | 5.20 | 83.10 |
| 2016 | 31.59 ± 28.97 | 5.10 | 200.90 | 22.96 ± 18.65 | 6.40 | 161.60 |
| 2017 | 30.43 ± 26.23 | 4.40 | 149.90 | 25.05 ± 26.27 | 6.50 | 216.40 |
| 2018 | 31.37 ± 21.74 | 5.40 | 134.80 | 25.23 ± 23.55 | 7.50 | 239.40 |
| 2019 | 25.01 ± 16.73 | 4.70 | 144.20 | 17.97 ± 12.66 | 4.90 | 149.60 |

**Table 3.** Average annual values of $SO_2$ in the studied health resorts in 2015–2019.

| Health's Resort | Ciechocinek | | | Sopot | | |
|---|---|---|---|---|---|---|
| **Year** | **Mean ± SD** | **Min** | **Max** | **Mean ± SD** | **Min** | **Max** |
| 2015 | 3.75 ± 3.39 | 0.40 | 20.40 | 1.97 ± 1.29 | 0.90 | 11.40 |
| 2016 | 3.02 ± 2.88 | 0.30 | 26.00 | 1.96 ± 1.97 | 0.80 | 22.30 |
| 2017 | 2.62 ± 3.49 | 0.30 | 29.30 | 1.61 ± 1.98 | 0.50 | 22.30 |
| 2018 | 2.09 ± 1.93 | 0.20 | 12.00 | 1.67 ± 1.45 | 0.70 | 10.20 |
| 2019 | 1.54 ± 1.30 | 0.20 | 10.40 | 1.36 ± 1.04 | 0.60 | 8.90 |
| **Health's Resort** | **Rabka Zdrój** | | | **Ustroń** | | |
| 2015 | 18.59 ± 11.47 | 3.01 | 71.61 | 7.15 ± 4.19 | 1.30 | 24.40 |
| 2016 | 15.56 ± 11.90 | 3.42 | 74.90 | 6.07 ± 4.61 | 1.30 | 31.90 |
| 2017 | 20.45 ± 15.25 | 3.91 | 89.75 | 7.53 ± 8.96 | 1.30 | 65.10 |
| 2018 | 20.44 ± 12.01 | 2.75 | 77.99 | 6.65 ± 6.60 | 1.80 | 51.40 |
| 2019 | 16.37 ± 11.16 | 2.85 | 64.27 | 5.52 ± 3.35 | 2.20 | 29.50 |

**Table 4.** Average annual values of $NO_2$ in the studied health resorts in 2015–2019.

| Health's Resort | Ciechocinek | | | Sopot | | |
|---|---|---|---|---|---|---|
| **Year** | **Mean ± SD** | **Min** | **Max** | **Mean ± SD** | **Min** | **Max** |
| 2015 | 13.75 ± 8.09 | 2.65 | 59.57 | 12.37 ± 7.64 | 2.89 | 53.38 |
| 2016 | 14.47 ± 9.76 | 2.01 | 51.02 | 13.25 ± 7.00 | 2.66 | 39.73 |
| 2017 | 11.26 ± 6.56 | 2.30 | 41.13 | 12.69 ± 7.66 | 3.09 | 59.20 |
| 2018 | 9.79 ± 5.27 | 1.57 | 31.18 | 14.64 ± 8.07 | 3.11 | 49.46 |
| 2019 | 9.62 ± 4.76 | 2.03 | 26.87 | 12.58 ± 5.98 | 2.95 | 36.04 |
| **Health's Resort** | **Rabka Zdrój** | | | **Ustroń** | | |
| 2015 | 9.80 ± 8.09 | 1.50 | 56.70 | 13.02 ± 6.26 | 3.73 | 45.90 |
| 2016 | 11.60 ± 12.15 | 2.00 | 74.40 | 12.88 ± 6.96 | 2.89 | 39.25 |
| 2017 | 10.77 ± 12.33 | 1.60 | 71.90 | 15.37 ± 10.79 | 3.66 | 80.61 |
| 2018 | 9.04 ± 7.86 | 1.70 | 45.40 | 14.49 ± 9.66 | 2.02 | 88.22 |
| 2019 | 7.76 ± 5.58 | 1.90 | 36.60 | 11.90 ± 6.89 | 2.97 | 53.38 |

The following are linear figures that show the average air temperature, relative humidity, wind speed, and visibility in 2015–2019 in four health resorts analyzed, such as Ciechocinek, Sopot, Rabka Zdrój, and Ustroń.

In 2015–2019, the increase in annual air temperature occurred in all health resorts. In the health resorts of Sopot, the average air temperature rises by as much as 0.95 °C in five years. Sopot is located beneath the surface of the ocean air flow from the west, and their impact from the east is low. The proximity of the sea, and especially the depths of Gdansk, influenced the thermal and humid phenomena of the Middle Ages, as well as the diverse geographical environment, especially the diverse topography. The lowest average annual temperatures were recorded in Rabka Zdrój. In 2015, the average annual currency price was 7.24 °C (Figure 2), and as much as 7.50 °C in 2019. In the years 2015–2019, the air temperature is 0.26 °C. Climate change is important for mountainous areas. Mountain ecosystems are highly vulnerable and vulnerable to climate change [32]. The highest average temperature in 2015–2019 in Ciechocinek is 10.53 °C. In the health resorts of Ustroń, in 2015, the average annual temperature was 10.26 °C, in 2016 and 2017,

the temperature dropped by about 1 C, and in 2018, there was an increase in the average annual air temperature.

**Table 5.** Average annual values of $B_{(a)}P$ in $PM_{10}$ in the studied health resorts in 2015–2019.

| Health's Resort | Ciechocinek | | | Sopot | | |
|---|---|---|---|---|---|---|
| Year | Mean ± SD | Min | Max | Mean ± SD | Min | Max |
| 2015 | 4.18 ± 3.40 | 0.10 | 12.20 | 9.58 ± 7.90 | 0.20 | 29.00 |
| 2016 | 3.58 ± 4.13 | 0.10 | 18.50 | 10.16 ± 11.86 | 0.20 | 40.00 |
| 2017 | 2.80 ± 3.24 | 0.10 | 16.40 | 6.81 ± 7.10 | 0.10 | 28.00 |
| 2018 | 3.17 ± 3.58 | 0.00 | 13.50 | 6.50 ± 6.82 | 0.20 | 23.00 |
| 2019 | 1.84 ± 2.19 | 0.00 | 10.00 | 3.50 ± 3.11 | 0.00 | 11.00 |
| Health's Resort | Rabka Zdrój | | | Ustroń | | |
| 2015 | 7.48 ± 7.62 | 0.50 | 25.70 | 5.40 ± 5.25 | 0.10 | 17.70 |
| 2016 | 6.20 ± 7.47 | 0.40 | 38.30 | 6.05 ± 8.00 | 0.20 | 42.90 |
| 2017 | 6.15 ± 7.60 | 0.40 | 30.30 | 7.09 ± 11.90 | 0.10 | 46.70 |
| 2018 | 7.55 ± 8.86 | 0.30 | 36.30 | 4.46 ± 5.46 | 0.20 | 20.10 |
| 2019 | 5.74 ± 6.41 | 0.20 | 23.40 | 3.52 ± 3.87 | 0.10 | 17.70 |

Legend, Mean ± SD—mean pollutant concentration ± standard deviation expressed in $\mu g/m^3$, Min—minimum concentration of pollutant expressed in $\mu g/m^3$, Mix—maximum concentration of pollutant expressed in $\mu g/m^3$.

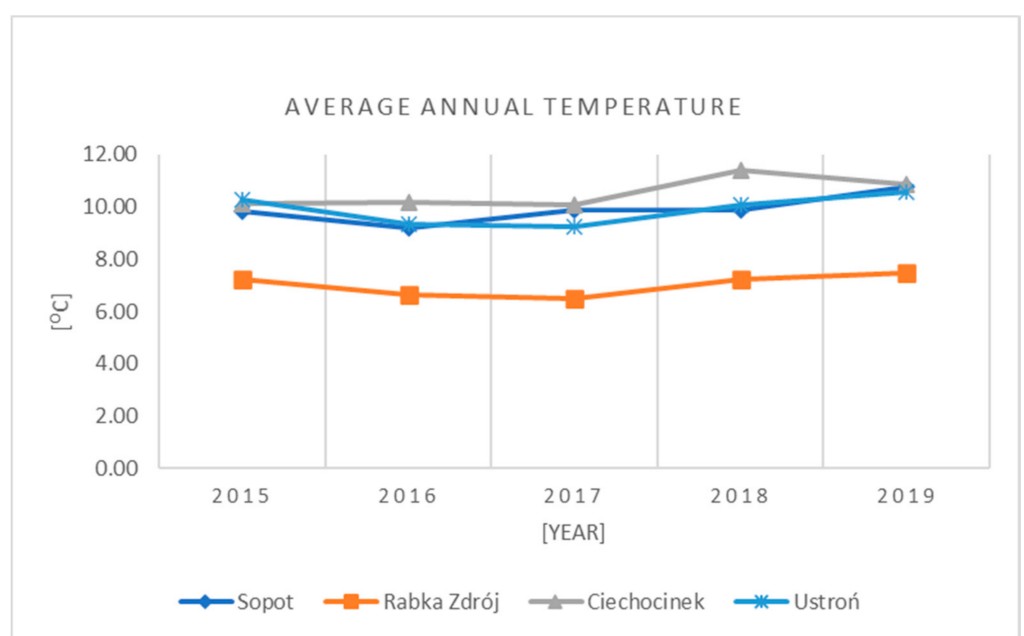

**Figure 2.** Average annual air temperature statistics for health resorts in 2015–2019.

The highest average wind speed(Figure 3) of the five years analyzed occurs in the health resorts of Ustroń and is 10.84 km/h, then in Sopot where the average wind speed in the health resorts during the period considered was 9.92 km/h consecutively, in Ciechocinek, 8.67 km/h and Rabka Zdrój, 5.54 km/h. Wind speed has a stimulating effect on mixing processes, and in addition, in the case of large urban agglomerations, which are a cluster of age emitters, it has an impact on the displacement of pollutants outside the city. Therefore, there is usually a decrease in the height of concentrations of pollutants with an increase in wind speed [33].

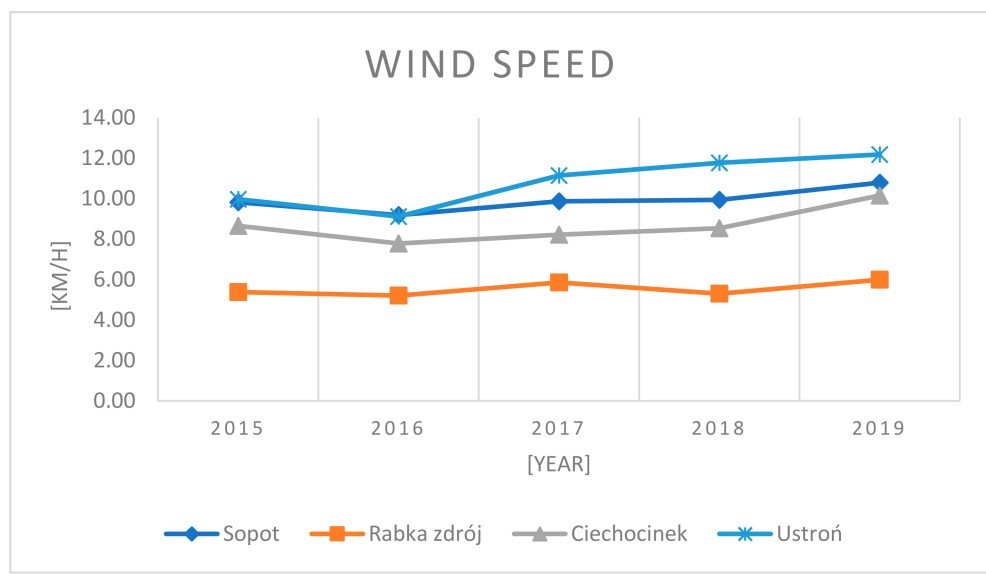

**Figure 3.** Average annual wind speed for health resorts communes in 2015–2019.

The highest average relative humidity (Figure 4) of 2015–2019 during the analysis period is 77.30% and occurs in Sopot. In Rabka Zdrój the average relative humidity is 76.61%, in Ciechocinek, 75.64%, In the health resort of Ustroń the average relative air humidity is 73.50% The development of basic statistics on meteorological elements is very important in order to determine their impact on concentration levels of harmful air pollutants. This is confirmed by numerous research works carried out by other researchers [34,35]. The relative humidity and air temperature [34] have the greatest impact on the con-centration of sulfur dioxide and particulate matter among meteorological elements. The level of particulate matter pollution of the atmosphere largely depends on meteor-ological conditions, primarily the direction and speed of wind, relative humidity of air, the intensity of solar radiation [35].

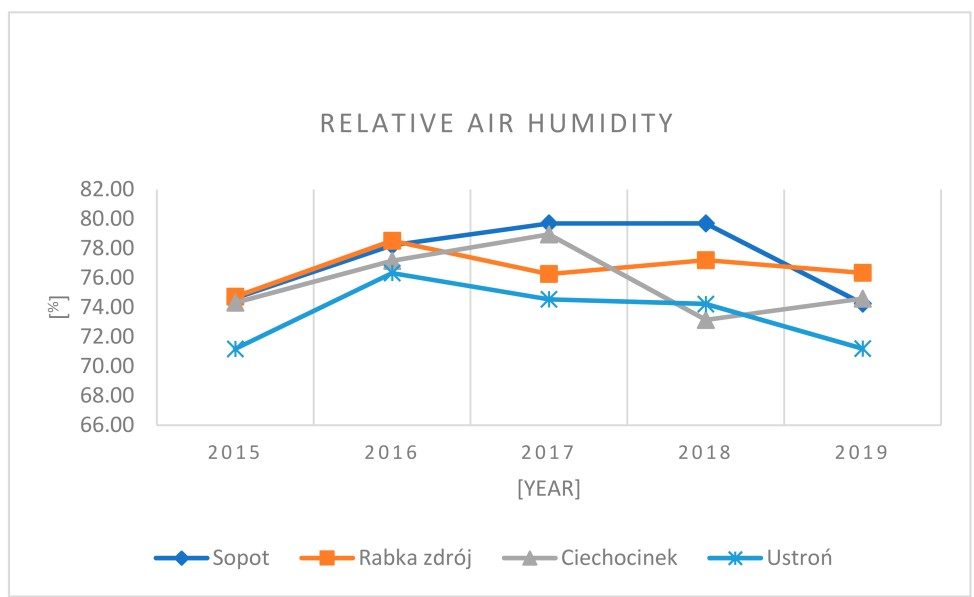

**Figure 4.** Average annual relative air humidity for health resorts in 2015–2019.

The average visibility during the period considered in the health resorts of Ciechocinek was 20.96 km, and in Rabka Zdrój, 20.35 km (Figure 5). In 2018, these health resorts saw a positive growing trend of visibility, which may indicate a decrease in concentrations of harmful air pollutants. A sudden decrease in visibility was noticed in Sopot; in 2017,

visibility was 17.92 km, and in 2018, 8.60 km. Such a rapid change in the range of visibility is a very unfavorable situation, which requires determining the reason for such sudden changes, identifying the source of the problem in order to eliminate it. In the health resorts of Ustroń, the average annual visibility in 2015 is 9.82 km, in 2016 and 2017, there is a gradual decrease in the range of visibility, and in 2018, you can see a slow beginning of a growing trend, which can testify, among other things, to the reduction of levels of harmful pollutants. Visibility is a very complex issue—it is directly due to the level of anthropogenic air pollution, but it is also shaped by meteorological conditions. The impact of anthropogenic air pollution on human health and visibility has been studied for decades. A number of studies have been carried out, not only to assess the benefits to human health of reducing air pollutant emissions [36], but also to understand how air pollution negatively affects visibility. In general, visibility is a good indicator of the degree of air pollution, and can also be used as a substitute for an assessment of the impact on human health [37].

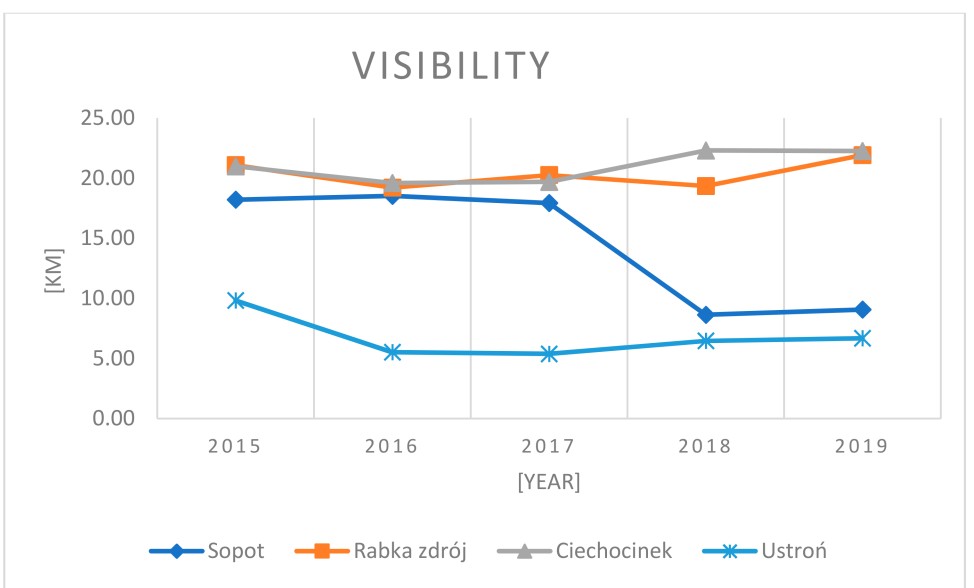

**Figure 5.** Average annual visibility for health resorts communes in 2015–2019.

*3.2. Analysis of the Occurrence of Episodes of Elevated $PM_{10}$ Concentrations in Health Resorts in Poland*

Table 6, attached below, shows the thresholds for exceeding the limit values for concentrations of pollutants. Within 24 h, the overrun value in accordance with the Regulation may not exceed 50 $\mu g/m^3$, and the allowable frequency of exceeding the allowable level in a calendar year shall be 35 days [28]. The second threshold > 100 $\mu g/m^3$ is an alert threshold that calls on state institutions to inform the public about an episode of elevated concentrations of pollutants.

**Table 6.** The number of days with exceedances of the limit value $PM_{10}$ (L > 50) and (L > 100) in the health resort surveyed in 2015–2019.

| Health Resorts | L > 50 [$\mu g/m^3$] | | | | | L > 100 [$\mu g/m^3$] | | | | |
|---|---|---|---|---|---|---|---|---|---|---|
| | 2015 | 2016 | 2017 | 2018 | 2019 | 2015 | 2016 | 2017 | 2018 | 2019 |
| Ciechocinek | **42** | 30 | 20 | **38** | 11 | 2 | 0 | 7 | 0 | 0 |
| Rabka Zdrój | **67** | **39** | **41** | **51** | 27 | 4 | 15 | 14 | 6 | 1 |
| Sopot | 2 | 3 | 10 | 14 | 1 | 0 | 0 | 2 | 0 | 0 |
| Ustroń | 16 | 19 | 21 | 26 | 5 | 0 | 3 | 11 | 6 | 1 |

The years in which the standards set out in Regulation [28] have been exceeded shall be indicated in bold.

Episodes of elevated concentrations are an atmospheric phenomenon resulting from the primary emission of particulate matter and gaseous pollutants into the air and the formation of secondary particulate matter as a result of chemical reactions occurring in the atmosphere, under meteorological conditions conducive to the accumulation of pollutants. Children, the elderly, and people already suffering from respiratory or blood diseases are particularly vulnerable to diseases associated with exposure to harmful airborne agents. Inhalation of air with a high content of harmful substances causes health problems, and in an increasing number of cases in the long term even contributes to death. The occurrence of episodes of high concentrations of particulate matter results in an increase in disease symptoms, mainly respiratory and vascular-blood diseases [1]. An episode of elevated concentrations of pollutants is a situation of one- or several days of elevated concentrations of $PM_{10}$ particulate matter, in which there is a significant exceedance of the daily limit for $PM_{10}$ (50 $\mu g/m^3$). The most common episodes of elevated concentrations of $PM_{10}$ particulate matter occur in Rabka Zdrój, a mountain health resorts located in the south of the Lesser Poland province. In accordance with the Regulation of the Minister of 24 August 2012 on the levels of certain substances in the air, the permissible frequency of exceeding the limit in a calendar year should not exceed 35 days. In 2015, 67 days out of 365, in 2016—39 days, in 2017—41, and in 2018—51 days is characterized by exceeding the average value of impurities for $PM_{10}$ particulate matter (Table 6). In 2016 (15 days) and 2017 (14 days), there were numerous exceedances above 100 $\mu g/m^3$. In 2018, the analysis showed a decrease in the frequency of crossings, below the standard set out in the Regulation. Short-term episodes of elevated concentrations of pollutants are mainly caused by low emissions. The problem of low emissions affects many communities, negatively affecting health. At the same time, this factor generates an increase in external costs for the health care of people with respiratory diseases. The increased impact of air pollution can be observed in urban areas, where increasing and compact housing makes it difficult to exchange air in cities and exacerbates the phenomenon of smog. Locally, this is also associated with bad weather and topographic conditions of cities [11]. Air pollution also affects flora, fauna, and cultural heritage. Monuments located in contaminated areas are destroyed and degraded at a much faster rate. A side effect of low emissions is acid rains, which cause, i. a., acidification of the soil, disruption of reproduction of living organisms, as well as damage to needles and leaves of trees [38]. However, it is promising to reduce the frequency of exceeding the daily limit value. In 2019, 27 days were characterized by exceeding the daily average value and let us say that this declining trend will continue. The situation is also unfavorable in Ciechocinek, a health resort located in central Poland. In 2015, the concentration levels of harmful pollutants exceeded the daily norm in 42 days, and 38 days in 2018. In 2017, although 24-h concentrations of particulate matter were exceeded 20 times as many as 7 days out of 20, the levels of harmful pollutants were more than 100 $\mu g/m^3$. In Ustroń, the highest number of days during which values exceeded the concentrations of pollutants allowed in the Regulation is 26 in 2018. In 2019, only 5 days were found in which levels of harmful pollutants exceeded the threshold of 50 $\mu g/m^3$. The best of all analyzed health resort is in the seaside town of Sopot. In 2015–2019, the permissible frequency of exceeding the permissible level did not exceed the 35-day standard, during which the achievable level was 50 $\mu g/m^3$.

### 3.3. Analysis of the Correlation between Harmful Air Pollutants Together with Benzo(a)pyrene and Meteorological Conditions

We calculated Pearson's correlation coefficient between meteorological conditions and air quality parameters: $PM_{10}$, $SO_2$, $NO_2$, and $B_{(a)}P$ in $PM_{10}$ [$\mu g/m^3$]. The correlation was determined between daily average values of the parameter. All the analyses were performed at the significance level $p < 0.05$. Table 7 shows all the data to demonstrate the significant as well as non-significant correlation of the variables. The red digits in the tables indicate the statistical significance of the correlation at 0.05.

**Table 7.** Analysis of the linear correlation (Pearson correlation coefficient) between air pollution with selected meteorological parameters and visibility in the health resorts of Ciechocinek, Ustroń, Rabka Zdrój, and Sopot in the years 2015–2019, the red digits indicate the significant correlations at $p < 0.05$.

| | Ustroń | | | | Sopot | | | |
|---|---|---|---|---|---|---|---|---|
| Variable | $T_{av}$ [°C] | Relative air humidity [%] | Wind speed [km/h] | Visibility [km] | $T_{av}$ [°C] | Relative air humidity [%] | Wind speed [km/h] | Visibility [km] |
| $PM_{10}$ [$\mu g/m^3$] | −0.379 | 0.104 | −0.247 | −0.028 | −0.103 | 0.071 | −0.388 | −0.382 |
| $NO_2$ [$\mu g/m^3$] | −0.476 | 0.228 | −0.078 | −0.141 | −0.455 | 0.128 | −0.221 | −0.162 |
| $SO_2$ [$\mu g/m^3$] | −0.577 | 0.139 | −0.054 | −0.080 | −0.362 | 0.170 | −0.385 | −0.367 |
| $B_{(a)}P$ in $PM_{10}$ [$\mu g/m^3$] | −0.714 | 0.261 | 0.015 | −0.163 | −0.693 | 0.386 | −0.135 | −0.188 |
| | Rabka Zdrój | | | | Ciechocinek | | | |
| Variable | $T_{av}$ [°C] | Relative air humidity [%] | Wind speed [km/h] | Visibility [km] | $T_{av}$ [°C] | Relative air humidity [%] | Wind speed [km/h] | Visibility [km] |
| $PM_{10}$ [$\mu g/m^3$] | −0.529 | 0.097 | −0.203 | −0.354 | −0.352 | 0.187 | −0.181 | −0.489 |
| $NO_2$ [$\mu g/m^3$] | −0.515 | 0.105 | −0.195 | −0.328 | −0.393 | 0.285 | −0.078 | −0.371 |
| $SO_2$ [$\mu g/m^3$] | −0.643 | 0.131 | −0.160 | −0.382 | −0.406 | 0.162 | −0.043 | −0.374 |
| $B_{(a)}P$ in $PM_{10}$ [$\mu g/m^3$] | −0.759 | 0.238 | −0.027 | −0.436 | −0.656 | 0.449 | −0.023 | −0.592 |

Table 7 of the correlation clearly indicates the relationship of average air temperature ($T_{av}$) to levels of harmful pollutants such as $PM_{10}$, $NO_2$, $SO_2$, and to the $B_{(a)}P$ content of $PM_{10}$. Correlation analyses confirm the influence of air temperature on the formation of concentrations of pollutants. Kobus and his co-authors came to similar conclusions in 2020. When comparing the correlation between average daily $PM_{10}$ concentrations and the daily values of selected meteorological indicators in the health resorts and agglomerations studied in 2019, the highest negative correlation with the minimum daily temperature is clearly observed. It is the strongest in the areas of health resorts and agglomerations located in the southern part of the country [22]. The strongest air temperature correlates with air pollution in Rabka Zdrój, which shows the lowest average annual air temperatures (Figure 2). In Ciechocinek, where air temperatures reached their highest values, it has been proven that as air temperatures rise, the impact on pollution decreases. With each of the municipalities with health resorts status, the impact of relative air humidity on air quality has been proven. The analysis showed the greatest effect on $B_{(a)}P$ levels in $PM_{10}$ has relative air humidity. In addition, the increase in the amount of PAHs in a given urban area is facilitated by an increase in relative humidity and strong winds blowing from directions where transport routes with significant vehicle traffic intensity are located [39] Low wind speeds increase levels of harmful pollutants. According to a 2013 Zaho study, wind speed was a major meteorological factor affecting both the visibility and concentration of particle mass. Low wind speeds combined with temperature inversions indicated stable meteorological conditions during the pollution process, which reduced the dispersion of pollutants and may have caused higher PM concentrations and lower visibility [40]. A negative correlation was noted between the wind speed in Sopot (Figure 3) and the level of harmful particulate matter $PM_{10}$ and $SO_2$. A negative correlation was observed in the case wind speed, when the conditions of dispersion of pollutants are improved [41]. Based on the analysis, visibility has been shown to be strongly correlated with air pollution. Emissions can cause impaired visibility, which makes visibility an important indicator of particulate matter pollution [42,43]. Therefore, the long-term trend in visibility may indicate a change in the state of air pollution [44,45]. Due to the rapid development of in

particulate matter and economic growth, people have incurred huge environmental costs for serious air pollution problems that can seriously harm public health [46]. In Sopot, on the basis of Tables 2–5, the lowest concentrations of impurities were shown, compared to the rest of the health resorts studied. The correlation analysis shows that we cannot reject hypothesis that air pollutants affect visibility at $p < 0.05$. In Sopot, where air quality was best and episodes of elevated concentrations of pollutants were the rarest (Table 6), correlations despite the fact that they showed relevance are the weakest among the health resorts analyzed.

### 3.4. Regression Analysis

We decided to propose multilinear regression model to determine the visibility as a linear combination of the meteorological and air quality data. Table 8 presents the basic statistical characteristics of the fitted models together with regression equations for health resorts: Ciechocinek, Rabka Zdrój, Ustroń, and Sopot.

**Table 8.** Values of person correlation coefficient (R) and multiple determination factor ($R^2$) between air pollution including meteorological conditions and visibility from four health resorts in 2015–2019.

| Health Resorts | N | Model Regression | R | $R^2$ | $R^2$Corrected | F | *p*-Value | BS |
|---|---|---|---|---|---|---|---|---|
| Ciechocinek | 1681 | $Vis = (-0.183 \cdot PM_{10})$ $+ \left(-0.317 \cdot B_{(a)}P\right)$ $+ (0.221 \cdot T_{av})$ $+ (0.496 \cdot RH)$ $+ (0.204 \cdot W_s)$ $+ 59.835$ | 0.838 | 0.703 | 0.702 | 793.62 | 0.00 | 6.417 |
| Ustroń | 1339 | $Vis = (0.115 \cdot T_{av}) + (-0.129 \cdot RH)$ $+ 16.64$ | 0.404 | 0.163 | 0.162 | 130.95 | 0.00 | 4.992 |
| Rabka Zdrój | 1781 | $Vis = (-0.039 \cdot PM_{10}) + (0.476 \cdot T_{av})$ $+ (-0.359 \cdot RH)$ $+ (0.254 \cdot W_s)$ $+ 43.198$ | 0.707 | 0.500 | 0.499 | 445.03 | 0.00 | 7.72 |
| Sopot | 1583 | $Vis = (0.430 \cdot SO_2) + (-0.337 \cdot NO_2)$ $+ \left(0.089 \cdot B_{(a)}P\right)$ $+ (0.265 \cdot T_{av}) + 7.72$ | 0.428 | 0.184 | 0.181 | 71.13 | 0.00 | 6.497 |

All the meteorological parameters were tested for the normality according to [26]. Eight independent variables were used for regression analysis. All of the analyzed variables were statistically significant ($p < 0.05$). The analysis showed that the multilinear model fits the best the data collected in Ciechocinek (the Pearson coefficient of correlation is R = 0.838). The result obtained indicates the significant relationship between visibility and meteorological parameters and concentrations of harmful pollutants in Ciechocinek. Ustroń found a relationship at R = 0.404, in Rabka Zdrój had R = 0.707, and Sopot, R = 0.428. Step-by-step regression analysis allows us, at the initial analysis stage, to remove from the model those variables that are not affected by the model form, facilitating further analysis. Five of the eight variables got into the model. From harmful concentrations of pollutants, the main influence on the form of the model in Ciechocinek was a relative humidity of air (0.496 Table 8) and level $B_{(a)}P$ (−0.317). Among the health resorts analyzed, $B_{(a)}P$, only in Ciechocinek, had a limiting effect on visibility, and nowhere else appeared in the analyses. In the health resorts of Ustroń, visibility is influenced by only two variables such as average air temperature and relative humidity. In Rabka Zdrój, the visibility of four variables out of the eight analyzed was found to affect the visibility. Visibility is affected by $PM_{10}$ harmful pollutants, average air temperature, relative air humidity, and wind speed. In the other

stations where the data were analyzed, a link was also found between the level of harmful $PM_{10}$ pollutants and visibility, but it is not that large. The developed model for Rabka Zdrój shows a positive relationship of wind speed on horizontal visibility, increasing it through more efficient circulation of polluted air. In Sopot, in the seaside health resorts, $SO_2$ and $NO_2$ are the most affected by visibility. The number of cases, i.e., days where the data met the requirements for analysis, is 1681 days for Ciechocinek, 1339 days for Ustroń, 1781 days for Rabka Zdrój, and 1582 days in Sopot. The more days–number of cases, the more reliable the model form you get. Humidity has a negative impact on the form of the model, but with regard to the model form of the other stations, it is negligible. The best-fit empirical regression models have shown a negative impact on the visibility of $PM_{10}$, $SO_2$, $NO_2$, and in particular $PM_{10}$ [47]. Statistical analysis showed that regression models were well matched to the observed data. Regression analysis confirmed the hypothesis that there is a relationship between visibility and concentration levels, as well as meteorological parameters.

### 3.5. Excessive Cancer Risk

The results of the ECR analysis are presented in Table 9. In three spas, the excess CR results in a higher cancer risk due to $B_{(a)}P$ than the average risk for Poland. The only health resort where the carcinogenicity risk due to $B_{(a)}P$ is lower than the average for Poland is Ciechocinek.

**Table 9.** Calculated excessive cancer risk for four health resorts in Poland.

| Year | Ciechocinek | Rabka-Zdrój | Sopot | Ustroń |
|------|-------------|-------------|-------|--------|
| 2015 | −11% | 56% | 112% | 19% |
| 2016 | −26% | 24% | 108% | 22% |
| 2017 | −42% | 30% | 42% | 54% |
| 2018 | −26% | 77% | 51% | 3% |
| 2019 | −44% | 82% | 7% | 6% |

ECR in Rabka-Zdrój remains high, on the order of tens of percent, and there is no downward trend, while for Sopot, we observe a decline in ECR values of around 100%. The results from Ustroń, especially 2018 and 2019, suggest a significant decrease in ECR, but 2017 shows that it may not be significant. From the table above, we can see that the CR associated with $B_{(a)}P$ can be reduced by traveling to Ciechocinek. On the other hand, it should be calculated separately for each location. We show how many stations in Poland have lower annual average $B_{(a)}P$ concentrations than the studied health resort. The results in Table 10 show the percentage of air quality stations in Poland with respect to annual average concentration. For three sites outside Ciechocinek, the ECR due to $B_{(a)}P$ concentration is higher than half of the air quality stations. Rabka-Zdrój in 2018 and Sopot in 2015–2016 were in the 10% of sites with the highest CR due to $B_{(a)}P$. Use of the cancer risk assessment tool identifies health risks associated with stays at spas considered to be health-promoting. Exposure to benzo(a)pyrene compounds can cause cancer [48]. The results of the analysis confirm the validity of further research in this direction.

**Table 10.** Contribution of air quality stations with above-normal cancer risk lower than four health resorts in Poland.

| Year | Ciechocinek | Rabka-Zdrój | Sopot | Ustroń |
|------|-------------|-------------|-------|--------|
| 2015 | 51% | 81% | 96% | 72% |
| 2016 | 42% | 77% | 94% | 73% |
| 2017 | 26% | 78% | 81% | 86% |
| 2018 | 36% | 90% | 85% | 65% |
| 2019 | 19% | 87% | 65% | 60% |

## 4. Discussion

Extensive statistical analysis was carried out for selected four health centers such as Ciechocinek, Sopot, Rabka Zdrój, and Ustroń. The analysis showed the main impact of $PM_{10}$ on air quality in places considered health-promoting. What is promising, however, is that the levels of harmful pollutants $PM_{10}$, $NO_2$, and $SO_2$ are gradually falling. Due to the increased episodes occurring in recent times, POP (Air Protection Programs) is introduced in Poland. The practical implementation of these programs leads to a reduction in the level of harmful pollutants. In 2015–2019, $B_{(a)}P$ levels in $PM_{10}$ decreased. Despite the steady decline in $B_{(a)}P$ concentrations in the air observed since the 1990s, exceedances of the acceptable level (1 ng/m$^3$ of the recommended human health objective) have still been recorded in the predominant area of the country. The main reason for the high level of benzo(a)pyrene in $PM_{10}$ is low emissions from heating buildings by burning fossil fuels (wood, gas, coal, oil). Linking meteorological conditions to air pollution allows for a deeper understanding of the subject. For example, visibility in many countries is used as an indicator of the degree of air pollution and can also be used as a substitute for human health impact assessment, where basic monitoring is not possible. Further studies and analyses of meteorological conditions are well known and linked to climate change, especially in mountainous areas, as mountain ecosystems are highly vulnerable and vulnerable to climate change. Travel to spas for health reasons can be problematic, because air quality is not satisfactory, as studies have shown. Rabka Zdrój is a submontane municipality where, in addition to spa functions, it also performs a tourist function. Due to population density and increased transport, the risk of exposure to cancer is highest. Staying in Rabka Zdrój for spa purposes is not indicated for vulnerable people in the risk group. The development of tourism is linked to the environment, but this activity has an impact on the environment, and the resulting changes can be detrimental to activities. Tourism activities are seen as one of the main factors causing environmental damage [49]. The intensive energy consumption associated with the provision of travel services leads to the emission of large quantities of pollutants [44]. Tourism, which is both 'guilty' and 'disadvantaged', also becomes a 'victim' of the effects of environmental degradation, which reduce the tourist attractiveness of the destination and, consequently, the income derived from leisure or travel [50]. Measures to reduce the level of harmful contaminants will have an impact on the effefctiveness of treatment of patients in health centers. The article shows the significance of conducting reliable monitoring of air quality, where this is particularly important. In addition to health-promoting values, spas also combine tourist functions. A large number of people and motor vehicles, has a negative impact on the environment and reduces tourist and spa values. An example of municipalities that have lost their spa status is Zakopane. In Zakopane, when it comes to spas, there is definitely too much crowd. The term "spas" itself is associated primarily with "holy peace", slow time, rest, and relaxation

## 5. Conclusions

Based on the statistical analysis, significant conclusions were drawn for the selected four health care facilities.

- The air quality in the health resorts was mainly affected by the level of harmful $PM_{10}$.

- In Ciechocinek, Ustroń, and Rabka Zdrój, the level of $NO_2$ is gradually decreasing, only in Sopot does the level of harmful pollutants remain at the same level in the analyzed years 2015–2019, and no decreases were recorded.
- The highest level of $SO_2$ was recorded in Rabka Zdr.
- Between 2015 and 2019, there was a decrease in the levels of $B_{(a)}P$ in $PM_{10}$.
- The highest average wind speed for the past five years has been in the resorts of Ustroń, and it is 10.84 km/h, followed by Sopot, where the average wind speed in the resorts during the study period was 9.92 km/h, 8.67 km/h in Ciechocinek, and in Rabka Zdrój, 5.54 km/h.
- Average visibility in the examined period in the Ciechocinek health resorts was 20.96 km, in Rabka Zdrój, 20.35 km.
- Episodes of elevated pollutant concentrations in 2015–2019 occurred most frequently in Rabka Zdrój.
- Correlations showed the strongest relationship between air pollution and average air temperature.
- Regression analysis showed a greater influence on the visibility of air pollution than meteorological conditions.
- Cancer risk analysis showed that in order to reduce the possibility of getting cancer, one should go to the Ciechocinek health resort as often as possible.
- Due to population density and increased transportation, the risk of cancer exposure is highest. Staying in Rabka Zdrój for spa purposes is not advisable for vulnerable people in the risk group.

**Author Contributions:** Conceptualization G.M. and E.A., methodology J.S.B. and E.A.; software, E.A. validation, E.A., formal analysis E.A., J.S. and J.S.B.; resources, E.A.; data curation, E.A.; writing—original draft preparation, E.A.; writing—review and editing E.A., G.M. and E.A., supervision, E.A.; project administration, E.A. All authors have read and agreed to the published version of the manuscript.

**Funding:** This research received no external funding.

**Institutional Review Board Statement:** Not applicable.

**Informed Consent Statement:** Not applicable.

**Data Availability Statement:** Archived measurement data from the Measurement Data Bank http://powietrze.gios.gov.pl/pjp/archives, https://.ogimet.com/gsynres.phtml (accessed on 20 February 2021).

**Conflicts of Interest:** The authors declare no conflict of interest.

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
