# Peer review of "The Quality of Air in Polish Health Resorts with an Emphasis on Health on the Effects of Benzo(a)pyrene in 2015–2019"

_climate, doi:10.3390/cli9050074_

Round 1
Reviewer 1 Report
The present paper aims to analyses the influence of indoor ambience contaminants over possible health risks (cancer) in different Spas based on real stapled data and statistical analysis. The Idea is of interest and original, real data makes the paper more credible but there are different points that must be improved:
1. Figures format must be improved (its quality is not in agreement with the journal indications).
2. Some tables cannot be read, like table 8
3. The statistical analysis must employ inferential statistics to reach more depth conclusions.
4. There is not a conclusion section (it is of interest to highlight the main conclusions obtained in the discussion section)
Author Response
The correspondent would like to kindly thank the reviewer for the valuable comments and suggestions made in the manuscript. This is the first work of the author of correspondence and all the information he considers valuable and helpful in future scientific work.
The author included all the reviewer's suggestions in the manuscript you submitted. The author followed the magazine's guidelines for formatting engravings and tables. They have been corrected according to the style indicated. Statistical analysis has been expanded to draw more relevant conclusions. In accordance with the right recommendations, a native conclusion was added, which emphasized the main results of the work.

Reviewer 2 Report
This manuscript pretends to evaluate the quality of air in Polish spa places including also assessment of cancer exposure by inhalation of benzo(a)pyrene.
Some comments/suggestions are presented below and some other specific comments are in the manuscript:
The article has a poor English and needs some reorganization.
Statistical analysis should be improved and also the presentation of the results.

Author Response
The correspondent author would like to kindly thank the reviewer for the valuable comments and suggestions made in the manuscript. This is the first work of the author of the correspondence and all the information he considers valuable and helpful in future scientific work.
In the file below, the author referred to each comment in the pdf file that was attached to the review. The corrections were proposed to the manuscript.
- Line 13. In the abstract there is no refernce about the weather conditions and this is a topic widely studiet.
Comment on: In order to effectively address public health problems, it is also necessary to take into account meteorological conditions when analysing air quality. A detailed analysis of the impact of meteorological conditions (average air temperature, wind speed, relative humidity and visibility) on air quality, based on forecasts, will also help implement air quality plans and strengthen the control of harmful pollution levels. Actions aimed at reducing the level of harmful impurities will affect the effectiveness of treatment of patients in spas. The article shows the correct way to conduct reliable monitoring of air quality and meteorological conditions, where this is particularly important.
- Line 26. Are the conclusins of this work?
Comment on: The abstract describes preliminary short conclusions aimed at encouraging the reader to familiarize themselves with the results and the whole article.
- Line 38. The text that you present here is the same as in [22]:
Air quality is not only a factor that directly and indirectly affects health, but also an important factor in determining the quality of life (well-being). This is important both for the functioning of various human organs, as well as for mental health and well-being. Subjective assessment of the quality and standard of living of mental stress can be conditioned by both long-term and short-term exposure to air pollution [2, 3]
Comment on: The introduction to the manuscript was corrected
- Line 48. format 1) then (4) 'or " format 1),
Comment on: Fixed text formatting error in manuscript
- Line 79. what is PVA? PAs? PWA? PW?
Comment on: PVA, Pas, PWA and PW designates polycyclic aromatic hydrocarbons (PAHs). This error occurred while translating the text of the manuscript and has been corrected
- Line 96 ??
Comment on: Translations error a correction has been made to the manuscript
- Line 103. reference?? –
Comment on: Corrections made and manuscript reference added
- Line 112. references from where the data was collected
Comment on: An amendment to the manuscript has been made. The insternet page from which meteorological data has been downloaded has been added
- Line 114. refer Figure 1 that shows the location of the health resorts
Comment on: A correction was made to the manuscript
- Line 130.use the same order here and in the results presented
Comment on: A correction was made to the manuscript
- Line 135. explain better, for example, for the regression analysis which were the variables considered The assumptions for determining Pearson correlation and perform regression analysis are met? (Such as normality of the data)
Comment on: As suggested in the manuscript, the issue of normality of the data was clarified in the results.
- Line 139: refer this table before it appears, explaining the information included in it –
Comment on: A correction was made to the manuscript
- Line 141: It would be easier to read if the same sequence was used for all the SPA's, starting with the pollutants that are common to all of tchem
Comment on: - A correction was made to the manuscript.
- Line 150.This section is difficult to read... Check the english and format!
Comment on: The author raised the level of language and corrected the correction in the manuscript and work with the format text.
- Line 155.??
Comment on: Correction made in style and language manuscript
- Line 176. The tables only have the air pollutants This information, in my opinion, would be more readable if presented in charts with error bars or boxplots. The only relevant characteristics are the mean value and the standard deviation?
Comment on: Rightly, reviewers have noted insufficient statistics. In addition to the mean value and standard deviation, minimum and maximum values have been added, which better characterize spa places. Corrections have been made to the manuscript
- Line 184 (table 3) The standard deviation is equal to the mean value? in this case. Is a coincidence or there is some typing error? It happens more times.-
Comment on: The error occurred while completing the tables. When you following reviewers' suggestions and correcting the tables, the error was fixed.
- Line 195. reference is missing
Comment on: Corrections have been made to the manuscript
- Line 196. mean value of the year
Comment on: According to the Minister's Regulation of 24 August 2012 on the levels of certain substances in the air, during the calendar year the level of PM10 particulate matter should not exceed the standard of 40 μg/m3/year.
- Line 199. The reference to the pair protection programmes is vague and does not mention why it is mentioned.
Comment on: In Poland, air protection programmes have been running in recent years due to poor air quality. In many cities such as Ciechocinek, the practical implementation of these programs results in a reduction in the levels of harmful pollutants.
- Line 206. Have you statisticaly checked the decreasing trend?
Comment on: An analysis of the basic statistical characteristics of the measuring series was carried out and on the basis of them the author did not see changes over the years in the concentration levels of harmful pollutants
- Line 223. ???-
Comment on: Corrections have been made to the manuscript
- Line 229. explain PY (appears only once)
Comment on: PY means (PAHs). This error occurred while translating the text of the manuscript and has been corrected
- Line 239 (chart 1) Refer before it appears
Comment on: A fair note, added an introduction to the charts in the manuscript
- Line 240.The period analysed is very short to be said that there is a clear upward trend... and to afirm this is due to the global warming as stated in the discussion
Comment on: The right note ,the five years analysed are too short a time period to be able to clearly indicate trends. Amendments have been made in line with the reviewer's opinion in manuskrypt.
- Line 246 FIGURE 1
Comment on: Fixed an error in the manuscript as suggested by the reviewer (chart 1 not figure 1)
- Line 271.Which ones?
Comment on: "This is confirmed" sentence through numerous research work carried out by other researchers'' refers to the references in literature number [33] and [34].
Fixed manuscript with valuable reviewer tips
- Line 292. in my opinion, this is more adequate for introduction
Comment on: Of course, the suggestion is very correct. The author, we wanted to clearly emphasize the problem of air quality in health resorts. Visibility is an added issue to the work, and it was about adding a few sentences here.
- Line 301. The table needs a previous text explaining the reason for being presented. Explain what is the 50th and the 100th. L>50 and L>100 units ...
Comment on: The manuscript takes into account the reviewer's correct suggestion and explains what the thresholds are and where they came from
- Line 314. From the beginning of this paragraph until here it seems to me introduction instead of results
Comment on: Valuable note. The author wanted to introduce the reader with a few sentences of introduction concerning episodes of elevated concentrations of pollutants because episodes of elevated concentrations have a very negative impact on human health and despite the introduction of air protection programs still occur in Poland.
- Line 316. significant here means what
Comment on: Translation error, including the correction in the text
- Line 322. I suggest to mark in the table the values above the limit.
Comment on: The author marked the values exceeding the norm in red and explained them in detail in the manuscript
- Line 325.significant? Was a statistical test applied?
Comment on: Table 6 was prepared using the statistical program "Statistica". A static analysis was performed to determine the frequency of exceeding the norm of 24 hours during the year with two defined thresholds. This analysis allowed us to draw conclusions
- Line 335. ??
Comment on: Corrected manuscript
- Line 357 materiality???
Comment on: Corrected manuscript
- Line 360.%?
Comment on: Corrected manuscript
- Table 7. Why is atmosphere pressure included here and not previously in the charts presented? What is the relevance?
Comment on: Due to poor correlations, atmospheric pressure was removed from the analyses.
- LINE 372. REFERENCE
Comment on: In the given case, there is no reference -this is the author's request based on the performed analysis of the Persona correlation. It was only in the next sentence that the author of the study of Kobus and others [22]
- Line 383. WWA? where is it explained?
Comment on: WWA means designates polycyclic aromatic hydrocarbons (PAHs).
This error occurred while translating the text of the manuscript and has been corrected
- Line 413. The kind of regression should be in materials and methods Which reference states that one ir more accurate than the other?...
Comment on: Followed the suggestion and removed the sentence from here
- Line 421. ??
Comment on: Typo error. Variance not visible
- Line 425. EXPLAIN VARIANCE-
Comment on: Based on his own knowledge, the author explained what variance is in the manuscript
- Line 430. Were the independent variables normalized? The way regression was carried out needs to be explained in the section materials and methods-
Comment on: A correction has been made to the manuscript. This issue is discussed in the chapter materials and methods
- Line 443. Which were the requirements?
Comment on: The requirements to be met in regression analysis are, i.a, the completeness of the data. Statitisca, having a set of data in the course of performing regression analysis itself determines the quantity-in this case the day that are suitable for carrying out the analysis
- Line 471. Explani the relevance of this analysis
Comment on: Good suggestion, manuscript corrections made
- Line 477. Disscusions or Conclusions its a mix with discussion, conclusions and some information for the introduction
Comment on: The author of the manuscript corrected the work with valuable conclusions and the reviewer's suggestion. He divided into sections discussion and conclusions.
- Line 483. This should have been explained before...
Comment on: The error has been corrected in the manuscript

Round 2
Reviewer 1 Report
After a major revision, the paper was improved. Despite this, the paper format must be revised again carefully like, for instance, references 50 and 51. After this minor revision, the paper can be considered as adequate for publication.
Author Response
The authors of the manuscript would like to thank the reviewer for his accurate comments and suggestions, which greatly improved the quality of the paper. The authors, following the reviewer's advice, improved the style and language and formatted the paper to meet the requirements of the journal.
Reviewer 2 Report
Although, in my opinion, the manuscript has improved after revision, there are still some points that need correction/explanation:
1. I think English needs to be improved and there are also some typing errors.
Examples:
- Lines 151 and 152 – independent and dependent are exchanged
- Tables 2, 3, 4 and 5 – remove “Average” from the title
- Format of PM10 (appears PM10, PM10, or pm10)
2. I am not satisfied with the following answers:
- Line 135. explain better, for example, for the regression analysis which were the variables considered The assumptions for determining Pearson correlation and perform regression analysis are met? (Such as normality of the data)
Comment on: As suggested in the manuscript, the issue of normality of the data was clarified in the results.
I do not see this in the manuscript.
- Line 425. EXPLAIN VARIANCE-
Comment on: Based on his own knowledge, the author explained what variance is in the manuscript
I do not agree with the explanation.
“R-squared (R2) is a statistical measure that represents the proportion of the variance for a dependent variable that's explained by an independent variable or variables in a regression model. R-squared explains to what extent the variance of one variable explains the variance of the second variable. So, if the R2 of a model is 0.50, then approximately half of the observed variation can be explained by the model's inputs.”
- Line 430.Were the independent variables normalized? The way regression was carried out needs to be explained in the section materials and methods.
Comment on: A correction has been made to the manuscript. This issue is discussed in the chapter materials and methods
I do not see the answer to the question “Were the independent variables normalized?” in the manuscript.
Author Response
The authors of the manuscript would like to thank the reviewer for his accurate comments and suggestion, which improved the quality of the paper significantly. The authors improved the style and language, and formatted the paper to meet the requirements of the journal.
Examples:
- Lines 151 and 152 – independent and dependent are exchanged
Comment on: Good comment from the reviewer, manuscript error corrected
- Tables 2, 3, 4 and 5 – remove “Average” from the title
Comment on: The ,,Average’’ of the titles of graphs 2, 3, 5 has been removed
- Format of PM10 (appears PM10, PM10, or pm10)
Comment on: The notation of PM10, NO2 and SO2 was standardized in article
- I am not satisfied with the following answers:
Line 135. explain better, for example, for the regression analysis which were the variables considered The assumptions for determining Pearson correlation and perform regression analysis are met? (Such as normality of the data) Comment on: As suggested in the manuscript, the issue of normality of the data was clarified in the results.
I do not see this in the manuscript.
Comment on: The author followed the author's suggestion and made corrections to the manuscript Line 147-148.
- Line 425. EXPLAIN VARIANCE-
Comment on: Based on his own knowledge, the author explained what variance is in the manuscript
I do not agree with the explanation.
“R-squared (R2) is a statistical measure that represents the proportion of the variance for a dependent variable that's explained by an independent variable or variables in a regression model. R-squared explains to what extent the variance of one variable explains the variance of the second variable. So, if the R2 of a model is 0.50, then approximately half of the observed variation can be explained by the model's inputs.”
Comment on: Thank you for pointing it out. There was misunderstanding with variance of the model and variance of the data and we decided to rewrite completely this section focusing more on the Pearson coefficient of correlation. Line 448-458
Line 430.Were the independent variables normalized? The way regression was carried out needs to be explained in the section materials and methods.
Comment on: A correction has been made to the manuscript. This issue is discussed in the chapter materials and methods
I do not see the answer to the question “Were the independent variables normalized?” in the manuscript.
Comment on: We tested the normality of the data. We included information about it in both in materials and methods section (around line 147) and in Results and discussion (around line 449).